# Aryl Hydrocarbon Receptor Signalling in the Control of Gut Inflammation

**DOI:** 10.3390/ijms25084527

**Published:** 2024-04-20

**Authors:** Irene Marafini, Ivan Monteleone, Federica Laudisi, Giovanni Monteleone

**Affiliations:** 1Gastroenterology Unit, Policlinico Universitario Tor Vergata, 00133 Rome, Italy; irene.marafini@gmail.com; 2Department of Biomedicine and Prevention, University of “Tor Vergata”, 00133 Rome, Italy; ivan.monteleone@med.uniroma2.it; 3Department of Systems Medicine, University of “Tor Vergata”, 00133 Rome, Italy; federica.laudisi@uniroma2.it

**Keywords:** Crohn’s disease, ulcerative colitis, IBD, intestine, AHR

## Abstract

Aryl hydrocarbon receptor (AHR), a transcription factor activated by many natural and synthetic ligands, represents an important mediator of the interplay between the environment and the host’s immune responses. In a healthy gut, AHR activation promotes tolerogenic signals, which help maintain mucosal homeostasis. AHR expression is defective in the inflamed gut of patients with inflammatory bowel diseases (IBD), where decreased AHR signaling is supposed to contribute to amplifying the gut tissue’s destructive immune–inflammatory responses. We here review the evidence supporting the role of AHR in controlling the “physiological” intestinal inflammation and summarize the data about the therapeutic effects of AHR activators, both in preclinical mouse models of colitis and in patients with IBD.

## 1. Mechanisms Controlling Intestinal Homeostasis

The normal intestine is infiltrated with huge numbers of immune cells, and this is due to the myriad of microbial and dietary antigens that continuously stimulate the gut immune system. This state of “physiological”, low-grade inflammation does not result in overt mucosal damage, because several mechanisms, each with a specific function, contribute to turning down the activity of the immune cells [1,2,3,4]. The intestinal epithelium contains several cell types (i.e., enterocytes, Paneth cells, goblet cells, tuft and cup cells, microfold cells, and enterochromaffin cells) that generate a physical and functional barrier protecting the host from harmful molecules and microorganisms in the intestinal lumen, enabling the tolerance of commensal organisms, and optimizing the absorption of nutrients [5,6]. Some dietary and microbial antigens can cross the intestinal epithelium. This occurs, for example, in Peyer’s patches or follicles, where there is a specialized epithelium consisting of M cells that can transport luminal antigens to the mucosa-associated lymphoid tissue and activate an immune response [6]. In addition, cells of innate immunity such as dendritic cells (DCs) or macrophages can sample luminal antigens, with the extension of protrusions between epithelial cells [7]. In normal conditions, these cells have anti-inflammatory properties as they produce suppressive cytokines, such as interleukin (IL)-10 and transforming growth factor (TGF)-β1, which can directly suppress both innate and adaptive immune response and favor the differentiation and activity of regulatory T cells (Tregs) [8,9]. Additionally, activated effector T cells undergo apoptosis, a phenomenon that limits the pool of antigen-specific reactive cells [10,11].

Host genetics contribute to maintaining intestinal homeostasis as substantiated by the fact that mutations in genes encoding for regulatory molecules (i.e., IL-10 receptor) or proteins involved in the intestinal barrier function lead to monogenic forms of inflammatory bowel diseases (IBD) [12,13,14,15], a group of chronic inflammatory disorders that affect the intestine [15]. Similarly, dietary components (i.e., fibers) can be fermented by intestinal bacteria and act as a source of short-chain fatty acids that provide energy for colonocytes and preserve the integrity of the intestinal barrier [16,17]. On the other hand, many other environmental factors (e.g., infections, drugs, cigarette smoking, red meat) can break mucosal immune homeostasis and contribute to the development of IBD [18,19,20].

The mechanisms by which the gut immune system recognizes and responds adequately to luminal antigens are not fully understood, even though recent studies have shed light on the role that some transcription factors play in the control of mucosal immune responses. One such factor is the aryl hydrocarbon receptor (AHR), a transcription factor ubiquitously expressed in vertebrate cells. In this article, we will review the available evidence on the role of AHR in the gut in physiologic conditions and the contribution of AHR deficiency in the pathogenesis of IBD. We will also describe the results of preclinical work in mouse models of IBD-like colitis, and of clinical trials in which the therapeutic modulation of AHR function has been tested in IBD patients.

## 2. AHR Ligands

AHR is a member of the periodic circadian protein (PER)–AHR nuclear translocator (ARNT)–single-minded protein (SIM) superfamily of transcription factors, in which the PER-ARNT-SIM domain senses both endogenous factors (e.g., oxygen tension or redox potential) and exogenous factors (e.g., polyaromatic hydrocarbons and environmental toxins) [21]. In the absence of a ligand, AHR is located in the cytoplasm bound to actin filaments as an inactive complex with several chaperones. Upon ligand binding, AHR translocates into the nucleus, where it is released from the complex, heterodimerizes with ARNT, and binds genomic regions containing its binding motif [dioxin response element (DRE)], inducing the transcription of target genes such as CYP1A1, CYP1A2, CYP1B1, and AhR repressor (AhRR) [22,23,24]. AHR signaling is also regulated at additional levels, including the proteasomal degradation of AHR protein, ligand metabolism by CYP1A1, and AHR/ARNT complex disruption by AhRR [24,25,26,27]. For a better description of the mechanisms controlling the AHR pathway, the reader is directed towards recent reviews [28,29,30,31].

Classically known as a mediator of the toxicity of environmental pollutants, such as 2,3,7,8-tetrachlorodibenzo-p-dioxin, AHR can be activated by many other ligands, including environmental, dietary, and endogenous aromatic compounds. For example, vegetables in the diet, particularly those belonging to the cruciferous family (e.g., broccoli, cabbage, and brussels sprouts), are a major source of AHR agonists such as indole-3-acetate (I3A), indole-3-carbinol (I3C) and 3,3′-diindolylmethane [32]. I3C can be broken down by gastric acid, resulting in the production of 3,3′-diindolylmethane (DIM) and indole [3,2-b] carbazole (ICZ), other potent agonists of AHR [33,34]. AHR agonists can also derive from dietary tryptophan, a ubiquitous dietary amino acid that is metabolized by commensal microorganisms (e.g., *Lactobacillus reuteri*) [35,36].

## 3. The Role of AHR in the Maintenance of Gut Mucosal Homeostasis

### 3.1. Anti-Inflammatory Effects of AHR in the Intestine

AHR signaling is critical for maintaining mucosal homeostasis and this regulatory effect seems to be the result of the ability of AHR to control the function of both immune and non-immune cells. Mice fed a diet free of AHR ligands have reduced levels of fecal immunoglobulin (Ig) A, a phenomenon that has been associated with changes in the content of some species of the family Erysipelotrichaceae [37]. Orally administered I3A, a gut microbiota-derived tryptophan metabolite, alleviates liver steatosis and inflammation in a mouse model of diet-induced non-alcoholic fatty liver disease. Notably, I3A does not alter the intestinal microbiome, suggesting that the I3A’s beneficial effects likely reflect the metabolite’s direct actions on the liver [38]. In colitic mice, the administration of I3C triggers an IL-22-dependent mechanism that attenuates colonic inflammation, and prevents the microbial dysbiosis caused by colitis by increasing a subset of Gram-positive bacteria known to produce butyrate [39]. Moreover, the intestinal microbiota of mice deficient in caspase recruitment domain-containing protein-9, a susceptibility gene for IBD [40,41], fails to metabolize tryptophan into AHR ligands and this is associated with an increased susceptibility to colitis [42].

### 3.2. Epithelial Cells

Mice with selective deletion of AHR in intestinal epithelial cells (Vil1^Cre^Ahr^fl/fl^) are unable to control *Citrobacter rodentium* infection and exhibit defective signaling in the WNT-β-catenin and ubiquitin E3 ligase pathways, which results in the uncontrolled growth of intestinal stem cells and epithelial cell malignant transformation. In this model, the administration of AHR agonists suppresses tumor progression [43]. AHR sustains also the intestinal barrier function, given its ability to increase the expression of tight junction proteins and the production of defensins (i.e., REG3β and REG3γ) [44,45]. The exposure of intestinal epithelial cells to AHR ligands enhances the levels of the IL-10 receptor, the signaling of which, under normal conditions, sustains gut epithelial cell proliferation and barrier integrity [15,46,47,48].

### 3.3. Intra-Epithelial Lymphocytes

AHR signaling is needed for the maintenance of both TCRγδ intraepithelial lymphocytes (IELs) and innate lymphoid cells (ILCs) in the gut. IELs act as a first line of defense and promote epithelial barrier organization and wound repair [49]. AHR deficiency is associated with a marked reduction in the number of IELs and increased susceptibility to gut epithelial damage [50]. AHR can also reprogram IELs into immunoregulatory T cells. An example is represented by the differentiation of CD4 + CD8αα+ double-positive IELs, a subset of regulatory cells that originate from small intestinal CD4+ T cells following the down-regulation of the transcription factor Thpok [51]. *Lactobacillus reuteri* generates indole derivatives of tryptophan that promote both AHR activation and Thpok down-regulation in CD4+ T cells with consequent CD4 + CD8αα+ double-positive IEL differentiation [51]. In mice with dextran sulfate sodium (DSS)-induced colitis, the activation of AHR by 6-Formylindolo (3,2-b) carbazole (FICZ) attenuates the apoptotic rate of CD8αα + TCRαβ+ IELs [52], probably as a result of the increased expression of the IL-15 receptor on the membrane of such cells, which serves as a positive regulator of IEL survival [53,54].

### 3.4. Innate Lymphoid Cells

ILCs are another subset of immune cells acting as ‘pre-primed’ terminal effector cells that constitutively reside in barrier tissues and respond to alarmins and cytokine signals released following tissue damage [55]. In contrast to T cells, ILCs lack antigen-specific receptors and do not require de novo proliferation and polarization for cytokine production [55,56]. ILCs are divided into various subgroups that are defined by their master transcription factor usage and cytokine-producing capacity [57]. ILCs of the ILC22 type include NKp46(+) and lymphoid tissue-inducer (LTi)-like subsets that express AHR and protect the intestinal mucosa from infection by secreting IL-22 that acts on mucosal epithelial cells, inducing their survival, proliferation, and secretion of antimicrobial peptides [55,58,59,60]. In line with this, AHR-deficient mice exhibit a marked reduction in the number of ILC22 and the secretion of IL-22 and are unable to mount a protective response during *Citrobacter Rodentium* infection [61]*,* a mouse model that mimics human colitis induced by attaching and effacing enterotoxigenic *E. coli* [62]. AHR-deficient mice also lack post-natally imprinted cryptopatches (CP) and isolated lymphoid follicles (ILF), but not embryonically imprinted Peyer’s Patches (PP) [61]. It was also shown that AHR induces Notch, which is required for NKp46 + ILC, while LTi-like ILC, CP, and ILF are partially dependent on Notch signaling [61]. The NKp46 + ILCs are intact in germ-free mice, contradicting the possibility that the AHR ligands inducing the development of these cells derive from the action of the flora on foods. In contrast, the differentiation of NKp46 + ILCs likely relies on AHR activation induced either by endogenous molecules, such as the tryptophan catabolite kynurenine (Kyn), or natural ligands present in the diet [50]. AHR-dependent production of IL-22 by ILC22 is also positively regulated by IL-36 receptor (IL-36R) signaling. IL-36R-deficient mice exhibit a significant impairment in the expression of IL-22, increased intestinal damage, and failure to contain *Citrobacter rodentium* infection. Such defects in IL-36R-deficient mice are secondary to the reduced production of IL-23 and IL-6, two key IL-22 inducers in the early and late phases of infection, respectively [63].

In the small intestine, dietary AHR ligands are needed for the differentiation of a subset of regulatory eosinophils, which express the C-type lectin domain family 4, member a4 (Clec4a4), a C-type lectin inhibitory receptor specific for glycans. Mice with a selective deficiency of AHR in eosinophils exhibit an expansion of ILC of type 2 and clear *Nippostrongylus brasiliensis* infection more effectively than wild-type mice [64].

### 3.5. T Cells

AHR expression is differently regulated in the various subsets of T helper (Th) cells, with Th17 cells and regulatory T cells (Tregs) expressing the highest AHR levels [65,66]. There is also abundant evidence supporting the promoting effect of AHR activation on Tregs [67,68,69]. A recent study showed that *Porphyromonas gingivalis* (Pg), a Gram-negative anaerobic microbe implicated in the development of periodontitis, an inflammation of the oral cavity that can be associated with IBD [70,71,72], targets the gut microbiota and suppresses the pathway of linoleic acid (LA) [73]. LA functions as an AHR ligand that suppresses Th17 differentiation while promoting Treg differentiation. Consistently, restoring LA levels in colitis mice challenged with Pg decreases the Th17/Treg cell ratio in an AHR-dependent manner, thus leading to the attenuation of mucosal inflammation [73]. Dietary L-tryptophan converted by host IDO1/2 enzymes, but not by gut microbiota, induces G-protein coupled receptor (Gpr) 15 transcription preferentially in Tregs via the AHR, increases the colonic number of Tregs and attenuates the ongoing inflammation in a mouse colitis model generated by *Citrobacter rodentium* infection or DSS treatment [32]. Mechanistically, AHR directly binds to open chromatin regions of the Gpr15 locus, a phenomenon that can be further enhanced by Foxp3 and suppressed by RORγt [33]. In line with this, the loss of dietary tryptophan perturbs the interaction between the host and intestinal microbial communities, thus promoting the deprivation of bacterially derived tryptophan metabolites and reduced numbers of Tregs [74]. On the other hand, there is evidence that mice infected with the protist *Blastocystis* ST7 develop colitis characterized by the reduction of Tregs and simultaneous expansion of Th17 cells. These CD4+ T cell alterations rely on the tryptophan metabolite indole-3-acetaldehyde (I3AA) produced by the parasite. I3AA acts as an AHR inhibitor and reduces the Treg subset in vivo and Treg development in vitro by increasing the expression of Smad7, an inhibitor of TGF-β1 signaling [75].

### 3.6. Dendritic Cells

AHR activation is also critical for the function of DCs. Triggering AHR activation in DCs leads to the reduced expression of CD80, CD83, and CD86, and diminished synthesis of pro-inflammatory cytokines (i.e., IL-1β, IL-23, and IL-12). Moreover, following the activation of AHR by FICZ, DCs become more efficient inducers of Tregs, and the adoptive transfer of such regulatory DCs to mice with 2,4,6-trinitrobenzenesulfonic acid (TNBS)-induced colitis alleviates the severity of the ongoing inflammation [76]. Mice with a specific deletion of AHR in DCs are highly susceptible to DSS-induced colitis. Co-culturing AHR-deficient DCs with intestinal epithelial organoids results in an aberrant WNT pathway and a shortfall in the differentiation and proliferation of intestinal epithelial cells, raising the possibility that AHR signaling in DCs is needed to promote the production of factors that regulate intestinal epithelial cell biology [77]. *Dubosiella newyorkensis*, a murine commensal bacterium, and its human homologue *Clostridium innocuum*, induce the production of short-chain fatty acids, especially propionate and L-Lysine (Lys). Notably, Lys stimulates the differentiation of regulatory DCs by enhancing tryptophan catabolism towards the kyn pathway through the activation of the metabolic enzyme indoleamine-2,3-dioxygenase 1 (IDO1) in an AHR-dependent manner. Consequently, *Dubosiella newyorkensis* rebalances the Treg/Th17 responses and ameliorates mucosal barrier injury in the DSS-induced colitis model [78]. Exopolysaccharide (EPS), an active molecule produced by *Bacillus subtilis*, protects mice from *Citrobacter rodentium*-induced colitis by inducing, in a TLR4-dependent manner, anti-inflammatory M2 macrophages or inhibitory DCs [79]. Analysis of the signaling events downstream of TLR4 showed that EPS induces IDO in DCs, and the inhibition of T cell proliferation by IDO-expressing DCs utilizes the kyn/AHR circuit [80]. The Quitana’s group has recently described an integrated systems approach, combining publicly available databases, zebrafish chemical screens, machine learning, and mouse preclinical models to identify environmental factors that control intestinal inflammation. This approach showed that the herbicide propyzamide, which is used on fruits, vegetables, and in ornamental gardens, amplifies inflammation in the small intestine and colon induced by TNBS and anti-CD3 monoclonal antibodies, respectively, in zebrafish and mice. The evaluation of the mechanisms by which propyzamide enhances gut inflammation showed that the herbicide alters the intestinal microbiota even though intestinal dysbiosis was not involved in the inflammatory effect of propyzamide. In contrast, propyzamide reduces AHR activation in DCs, thus enhancing NF-κB activation and, hence, propagates pathogenic T-cell responses in the gut [81].

Collectively, the above findings indicate that AHR activation amplifies a multitude of counter-regulatory signals with the downstream effect of maintaining mucosal homeostasis in the gut (Figure 1).

## 4. AHR Down-Regulation in Crohn’s Disease Contributes to Sustaining the Mucosal Cytokine Response

Although the cause of Crohn’s disease (CD) and ulcerative colitis (UC), the main IBD in human beings, remains unknown, a large body of evidence suggests that these pathologies arise in genetically predisposed individuals as a result of the action of many environmental insults, which trigger an abnormal mucosal immune inflammatory response characterized by a massive production of inflammatory cytokines and a variety of defects in counterregulatory mechanisms [1,82,83,84,85]. In this context, we have demonstrated that the inflamed gut of CD patients contains reduced levels of AHR RNA transcripts, as compared with the uninflamed mucosa of the same patients and normal controls [86]. In contrast, no reduction in AHR RNA expression was seen in the inflamed colon of UC patients, as compared to normal controls, indicating that the down-regulation of AHR in CD does not rely simply on mucosal inflammation. Flow cytometry analysis of AHR-expressing immune cells in IBD mucosa revealed that, in CD, AHR expression is diminished in both CD4+ T cells and natural killer cells [86]. Functionally, activation of AHR in lamina propria mononuclear cells (LPMCs) isolated from the inflamed intestine of CD patients with FICZ increased the production of IL-22 and reduced the expression of interferon (IFN)-γ and T-bet. This finding was later confirmed by other authors showing that additional AHR ligands induce IL-22 production by T cells isolated from IBD patients [87]. AHR gene polymorphisms have been associated with an increased risk of developing IBD [35,36,37].

The mechanisms mediating the reduced expression of AHR in the intestine of CD patients are not fully elucidated. A possibility is that such an alteration relies on defects in the TGF-β1 signaling pathway [44], as TGF-β1 induces AHR in normal intestinal LPMC, while in CD LPMCs, which are characterized by a Smad7-dependent TGF-β 1 signaling inactivation [88,89], TGF-β1 does not enhance AHR expression unless Smad7 is knockdown [45]. These data were consistent with the demonstration that the intestinal T cells of Smad7-transgenic mice contain reduced levels of AHR and produce low amounts of IL-22 following FICZ stimulation. Moreover, in a model of colitis induced by the transfer of T cells into recombination-activating gene 1-deficient mice, the administration of FICZ to mice did not attenuate intestinal inflammation obtained with the transfer of Smad7 transgenic T cells [45]. There is also evidence that microRNA 124, which sustains IBD-like colitis in mice, can contribute to the reduced expression of AHR in intestinal epithelial cells of CD patients [41].

The CD-associated chronic inflammation can evolve into the formation of stenosis, which is the most frequent indication for surgery [90]. AHR is expressed by intestinal fibroblasts, and the activation of AHR by FICZ in CD intestinal fibroblasts reduces collagen production preliminary, thus suggesting the involvement of AHR in the fibrogenic processes [91].

## 5. AHR Signaling Attenuates Experimental Colitis

Several studies have convincingly shown that AHR activation is useful for limiting intestinal tissue-damaging pathogenic responses. In both TNBS-induced and T-cell transfer colitis models, the administration of FICZ to mice reduces the severity of colitis, dampens the production of Th1 cytokines, and increases IL-22 levels [86]. These AHR-driven regulatory effects are abrogated by the neutralization of the IL-22 supporting the role of this cytokine in the AHR-mediated immunosuppression. Similarly, 3,3′-diindolylmethane, a natural ligand of AHR, attenuates oxazolone-induced colitis [92]. AHR-deficient mice are more susceptible to DSS-induced colitis than wild-type mice, and AHR activation in such a model reduces the degree of inflammation [93]. In the same model, the activation of AHR by the administration of *Lactobacillus bulgaricus* OLL1181 inhibits ongoing colitis [94]. In a humanized murine model in which human CD4+ T cells drive colitis upon exposure to TNBS, the administration of the non-toxic AHR agonist 2-(10H-indole-30-carbonyl)-thiazole-4-carboxylic acid methyl ester (ITE) ameliorates colitis and increases the expression of CD39, Granzyme B, and IL10-secreting human Tregs [95]. In DSS-induced colitis, alpinetin, a flavonoid compound extracted from the seeds of the Alpinia katsumadai Hayata that acts as a strong AHR activator, alleviates colitis, and this effect is accompanied by a restored Th17/Treg balance in colons [96]. Moreover, in DSS-treated mice, the administration of Indigo naturalis (IN), a traditional Chinese medicine that contains ligands for AHR, increased Helios-positive Tregs and major histocompatibility complex class I- positive colonic epithelial cells [97].

## 6. Evidence Supporting the Regulatory Effect of AHR Pathway in Humans: Data from Clinical Studies

Considering the preclinical data outlined in the previous paragraphs and the important role of AHR in regulating pathological and physiological processes, AHR activators have been tested in humans for therapeutic purposes (Table 1). In the first multicenter, randomized, double-blind clinical trial using IN, 86 Japanese UC patients were enrolled [98]. Patients were randomized to receive IN at three different dosages or placebo (1:1:1:1) for 8 weeks. The primary endpoint was the rate of clinical response at week 8 (defined as a 3-point decrease in the Mayo score and a decrease of at least 30% from baseline, with a decrease of at least 1 point for the rectal bleeding subscore or absolute rectal bleeding score of 0–1). The main secondary endpoint was the rate of clinical remission at week 8; mucosal healing at week 8 was also assessed. The rates of clinical response at week 8 were significantly higher for patients receiving IN compared to the placebo, with a clear dose-dependent trend (placebo, 13.6%; 0.5 g IN, 69.6%; 1.0 g IN, 75.0%; 2.0 g IN, 81.0%). The clinical remission rate at 8 weeks was achieved significantly only in the two highest dosages of IN compared with the placebo, while the rates of mucosal healing were 13.6%, 56.5%, 60.0%, and 47.6%, respectively, for the placebo group and the three doses of IN [98]. Mild liver dysfunction was documented in 10–20% of patients who received 0.5–2.0 g of IN daily but spontaneously disappeared in the majority of these patients without dose adjustments. However, evidence about the possible development of pulmonary arterial hypertension in a patient who independently took IN for 6 months outside of this trial led to the discontinuation of this study and limited the large-scale, long-term use of this compound.

Later on, the same group tested the use of topical IN, in the form of suppositories, in UC patients with endoscopically active disease in the rectum and sigmoid colon [99]. This was a small open-label, single-center study in which 10 UC patients were prospectively enrolled. All patients received, once daily, 50 mg IN suppository for 4 weeks. The primary endpoint of this study was safety at week 4. Secondary endpoints were the rate of rectal bleeding subscore of 0, clinical remission, and mucosal healing after 4 weeks of treatment. Only one treatment-related adverse event, namely the development of anal pain, was recorded in one patient. At week 4, the rates of clinical remission and mucosal healing were 30 and 40%, respectively. Mayo rectal bleeding subscores significantly improved after treatment, but only in patients with a Mayo endoscopic subscore ≤ 2 and not in patients with a Mayo endoscopic subscore of 3 [99]. However, all considerations of the safety and efficacy of this formulation are limited by the low number of patients included in the study.

In another subsequent multicenter, randomized, placebo-controlled trial, the effect of oral IN in mild-to-moderate UC patients was tested [100]. Nineteen UC patients were assigned to the placebo group, and 23 were assigned to receive IN (500 mg) twice a day for two weeks. In the placebo group, no change in the Lichtiger index (an 8-item measure designed to assess disease activity in patients with UC [101]) was observed, while in the IN group, the Lichtiger index improved significantly. No significant adverse event related to treatment was recorded.

In a small open-label, dose-escalation study conducted in the United States, 11 patients with UC were treated either with IN 500 mg/day or 1.5 g/day for 8 weeks and subsequently followed up for a 4-week non-treatment period [102]. The primary endpoint was clinical response at week 8, assessed by the total Mayo score. Secondary endpoints included clinical remission, changes in Ulcerative Colitis Endoscopic Index of Severity, quality of life, C-reactive protein, and fecal calprotectin levels from baseline. In this study, AHR signaling was monitored by measuring RNA levels of CYP1A1, a downstream target of AHR activation. After 8 weeks of treatment, 10/11 patients achieved a clinical response, and 3 patients were in clinical remission. An improvement in endoscopic severity, biomarkers, and quality of life was observed in all patients. Following treatment cessation, 6 patients worsened, and four patients progressed to colectomy. RNA levels of CYP1A1 showed a 12,557-fold increase between baseline and week 8 in the colon tissue from six evaluated patients [102].

Recently, Ben-Horin and coworkers tested the efficacy of QingDai (a form of Indigo) combined with curcumin in the treatment of patients with active UC, based on preclinical and clinical data demonstrating the efficacy of both compounds separately in vivo [103]. This study was divided into two parts. Part I was an open-label trial in which the combination of curcumin and QingDai (CurQD) was given to 10 UC patients with a Simple Clinical Colitis Activity Index (SCCAI) score of 5 or higher, and active colonic inflammation (defined by a score of 2 or higher in the modified Mayo endoscopic subscore) for 4 weeks, which extended proximally to the rectum (>15 cm) at the screening colonoscopy. Part II was a double-blind, randomized, placebo-controlled induction trial in which CurQD was given for 8 weeks, with additional 8 weeks of maintenance treatment for responders. The same inclusion criteria were adopted for both parts of the study. The primary outcome of part I of the trial was the percentage of patients in clinical remission at week 4. The co-primary outcome of part II was the proportion of patients with clinical response (reduction in SCCAI of more than 3 points from baseline) and objective evidence of reduced inflammatory activity (Mayo score improvement ≥1 or 50% calprotectin reduction) at week 8.

In part I, 7 of 10 patients responded and 3 of 10 achieved clinical remission. Of the 42 patients in part II, 43% and 8% of CurQD and placebo patients, respectively, achieved the coprimary endpoint (*p* = 0.033). Clinical response was observed in 85.7% vs. 30.7% (*p* < 0.001) of patients and clinical remission in 14/28 (50%) vs. 1/13 (8%) in the CurQD and placebo groups, respectively [103]. An increased mucosal expression of CYP1A1 was documented in treated patients. Although the number of treated patients was limited, no case of pulmonary arterial hypertension was recorded.
ijms-25-04527-t001_Table 1Table 1Table summarizing clinical studies in which AHR activators were used.CompoundFormulationTarget PopulationType of StudyReferenceIndigo NaturalisCapsules  0.5–2.0 g per dayUlcerative colitismulticenter, randomized, double-blind clinical trial[98]Indigo NaturalisSuppositories  50 mg per dayUlcerative colitisopen-label, single-center study[99]Indigo NaturalisCapsules  1 g per dayUlcerative colitismulticenter, randomized,  placebo-controlled trial[100]Indigo NaturalisCapsules  500 mg–1.5 g/dayUlcerative colitisopen-label, dose-escalation study[102]QingDaiCapsules  Indigo Naturalis 1.5 g + Curcumin 500 mgUlcerative colitisPart 1: open-label trial  Part 2: double-blind, randomized,  placebo-controlled[103]


## 7. Conclusions and Future Directions

The data described in this article point to the role of the AHR as a keeper of the physical and immunological barriers present in the gut. Consistently, defects in AHR expression and/or activation are supposed to facilitate the propagation of pathogenic inflammation, which ultimately leads to tissue damage. This appears to be the case of IBD, in which the reduced expression of AHR in epithelial and immune cells contributes to promoting several pathways that alter the epithelial barrier integrity, expand the effector cytokine response, and block the function of anti-inflammatory molecules. Therefore, enhancing AHR activation could be a novel and promising way to dampen IBD-associated mucosal inflammation. In this context, many studies have documented the benefit of AHR activators in the preclinical models of IBD-like colitis, and some AHR-related compounds have already been tested with success in IBD patients. However, the limited number of patients enrolled in these studies reduces the relevance of the clinical data. Moreover, the clinical trials with natural AHR ligands have suffered a setback due to the possible development of serious side effects related to the use of these compounds. Many of the currently available natural AHR agonists are considered inadequate for clinical use due to low activity, inadequate pharmacokinetics, or toxicity. Therefore, in recent years, new synthetic agonists have been synthesized [87,104]. These compounds are more stable and have limited toxicities, due to their rapid clearance, than natural ligands, and exert desirable therapeutic effects [105]. Further experiments dedicated to studying their effects and improving their mode of delivery are, however, needed to possibly test these compounds in patients with IBD.

Many questions about the factors/mechanisms regulating AHR expression in the human gut, particularly in IBD, remain open. For instance, we do not yet know whether there is a cell-specific regulation of AHR in the gut and which factors, other than Smad7, account for the reduced expression of AHR in CD. Similarly, it remains unclear whether the defect in AHR content is limited to some of the evolutive phases of CD and whether all the phenotypes of the disease are marked by changes in AHR expression. Finally, further studies are needed to evaluate the therapeutic potential of AHR agonists in CD because, so far, the clinical trials have been conducted in UC but not in CD patients.

Because AHR activation triggers the differentiation of Tregs and the production of IL-22, it is conceivable that AHR agonists can have a place in the therapeutic armamentarium of other intestinal immune-mediated diseases. In this context, we have previously shown that AHR expression is reduced in the gut of patients with active celiac disease, and that the treatment of celiac disease IELs and LPMC with FICZ reduces the levels of inflammatory cytokines, granzyme B and perforin, as well as the fact that AHR activation protects mice against poly I:C-induced intestinal atrophy [106].

Notably, AhR-null mice do not exhibit an overt immunological phenotype and have no intestinal pathology [107,108]. This indicates clearly that the AHR deficiency documented in CD, as well as in celiac disease mucosa by itself, should not be sufficient to drive pathology. However, the exposure of AhR-deficient mice to inflammatory stimuli triggers detrimental immune responses in main barrier organs, including the lung, the skin, and the gastrointestinal tract. Studies aimed at evaluating how, in the absence of AHR, environmental risk factors amplify the mucosal inflammatory pathways could shed light on the pathogenesis of these diseases.

## Figures and Tables

**Figure 1 ijms-25-04527-f001:**
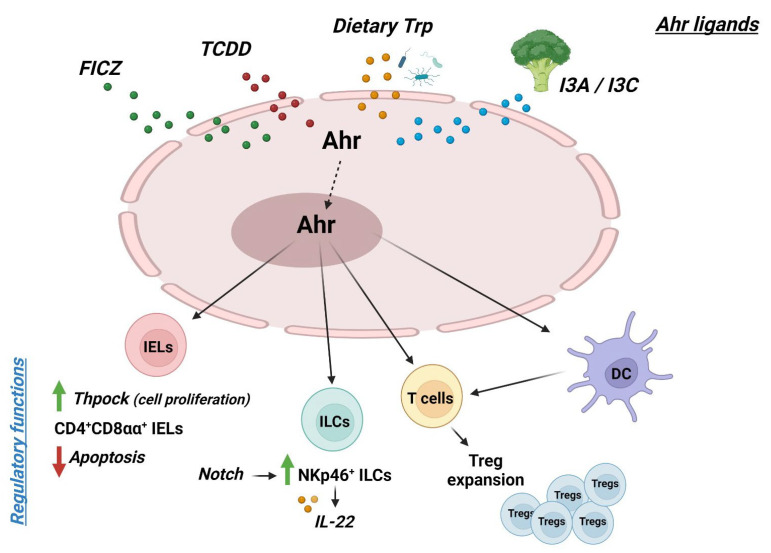
Schematic representation of the role of AHR in the maintenance of gut mucosal homeostasis; abbreviations: FICZ: 6-Formylindolo (3,2-b) carbazole; TCDD: 2,3,7,8-tetrachlorodibenzo-p-dioxin; Trp: tryptofhan; I3A: indole-3-acetate; I3C: indole-3-carbinol; Ahr: Aryl Hydrocarbon receptor; IELs: intraepithelial lymphocytes; ILCs: innate lymphoid cells; Treg: T regulatory cells; DC: dendritic cells; red arrow: decrease; green arrow: increase (figure created by Biorender.com; publication and licensing right of 25 March 2024).

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
