# Peer review of "Aryl Hydrocarbon Receptor Signalling in the Control of Gut Inflammation"

_ijms, 2024, doi:10.3390/ijms25084527_

Round 1
Reviewer 1 Report
Comments and Suggestions for Authors
The submitted manuscript is a short review on the role of AhR signalling in gut inflammation - an interesting topic, which has been a subject of a number of reviews recently.
However, in the present form it is not suitable for publication.
1) Paragraph 3 – the data presented in the paragraph is somewhat mixed up; therefore, please, arrange the content into separate fragments about
3.1. Deficient AhR signalling
3.2. Increased AhR signalling
2) Please, present the data in 3.1. and 3.2. keeping and order – animal studies first,
human studies second
3) please, add short paragraph summin up the anti-inflammatory activities of AhR
4) paragraph 4- Again, please present the data in the order – animal studies first, human studies second
5) Please, change the title of paragraph 7 into Conclusions and future directions; please specify what future studies should be undertaken
6) Last sentence of paragraph 7 mentions gluten involvement – there was not a single sentence about gluten earlier, please remove or add appropriate data, if relevant to AhR, into the main text
Comments on the Quality of English LanguageLanguage corrections are necessary. Below some examples of sentences that require editing
Line 22 – The normal intestine is infiltrated with huge numbers of immune cells and this is because
Line 65 - In the absence of a ligand, AHR is present in the cytoplasm bound to actin
82-83 AHR agonists can also derive from diet-derived
99-100 Moreover, the intestinal microbiota of mice knock-out for caspase recruitment domain-containing protein-9, a susceptibility gene for IBD [40,41], fails to metabolize tryptophan into AHR ligands and increases susceptibility to colitis – unclear meaning, what increases the susceptibility to colitis – microbiota?
Author Response
Editor in Chief
Manuscript title: “Aryl Hydrocarbon Receptor Signalling in the control of gut in-flammation”
Rome, 11/04/2014
Dear Editors,
thank you for your e-mail dated 10.04.2025, and for allowing us the opportunity to improve our manuscript. We would also like to thank the reviewers for their helpful comments and suggestions.
As indicated in the point-by-point reply to the reviewers’ comments below we have revised the manuscript taking into account all the issues raised by the reviewers. Since we hope these changes successfully address the reviewers’ comments, we would like to resubmit the review for publication.
Changes are underlined in the revised version of the manuscript.
Yours Sincerely,
Giovanni Monteleone (on behalf of all the authors)
Department of Systems Medicine
University of Rome Tor Vergata, Via Montpellier, 1, 00133 Rome, Italy
Phone +39.06.72596158
Fax +39.06.72596158
Email: gi.monteleone@med.uniorma2.it
Reviewer 1:
The submitted manuscript is a short review on the role of AhR signalling in gut inflammation - an interesting topic, which has been a subject of a number of reviews recently.
However, in the present form it is not suitable for publication.
1) Paragraph 3 – the data presented in the paragraph is somewhat mixed up; therefore, please, arrange the content into separate fragments about
3.1. Deficient AhR signalling
3.2. Increased AhR signalling
2) Please, present the data in 3.1. and 3.2. keeping and order – animal studies first,
human studies second
3) please, add short paragraph summing up the anti-inflammatory activities of AhR
4) paragraph 4- Again, please present the data in the order – animal studies first, human studies second
Response: We would like to thank the reviewer for her/his helpful suggestions. As suggested also by reviewer 2, we have rearranged the content and divided paragraph 3 into sub-paragraphs in order to facilitate the reader's understanding of this section. However, the order of the concepts has been left as in the original version of the paragraph, which is entirely devoted to pre-clinical evidence. Clinical data are all included in paragraph 6. There is also a separate paragraph devoted specifically to mouse evidence. The references to animal studies included in paragraph 3 are functional for highlighting the role of AHR in specific cell types/compartments. AHR has an anti-inflammatory/counter-regulatory role in all cell types and this is what the paragraph describes.
5) Please, change the title of paragraph 7 into Conclusions and future directions; please specify what future studies should be undertaken
Response: The title of paragraph 7 was changed according to the reviewer’s suggestion
6) Last sentence of paragraph 7 mentions gluten involvement – there was not a single sentence about gluten earlier, please remove or add appropriate data, if relevant to AhR, into the main text
Response: The word gluten was removed from the text.
Language corrections are necessary. Below some examples of sentences that require editing
Line 22 – The normal intestine is infiltrated with huge numbers of immune cells and this is because
Line 65 - In the absence of a ligand, AHR is present in the cytoplasm bound to actin
82-83 AHR agonists can also derive from diet-derived
99-100 Moreover, the intestinal microbiota of mice knock-out for caspase recruitment domain-containing protein-9, a susceptibility gene for IBD [40,41], fails to metabolize tryptophan into AHR ligands and increases susceptibility to colitis – unclear meaning, what increases the susceptibility to colitis – microbiota?
Response: All the indicated sentences were edited according to the reviewer’s suggestion and the whole text was revised to detect grammatical errors.
Reviewer 2 Report
Comments and Suggestions for Authors
It is an in depth analysis of the role of the Aryl Hydrocarbon Receptor [AHR] [a] in controlling the "healthy" intestine inflammation, [b] as a "treatment" in experimental colitis in mice and [c] in the early steps in IBD patients.
This is a review successfully gathered all information regarding the role of AHR in experimental and clinical colitis.
This review is well structured and contains the up-to date literature.
There is one table, summarizing the current clinical research and one, well designed figure depicting the way of action.
My unique suggestion is that the authors should divide chapter 3, on "the role of AHR in the maintain ace of gut homeostasis", into sub-part, according for example it effect on the different cell types; the only reason being to be easily understandable, since it is too long - 4 page, including the figure.
Author Response
Editor in Chief
Manuscript title: “Aryl Hydrocarbon Receptor Signalling in the control of gut in-flammation”
Rome, 11/04/2014
Dear Editors,
thank you for your e-mail dated 10.04.2025, and for allowing us the opportunity to improve our manuscript. We would also like to thank the reviewers for their helpful comments and suggestions.
As indicated in the point-by-point reply to the reviewers’ comments below we have revised the manuscript taking into account all the issues raised by the reviewers. Since we hope these changes successfully address the reviewers’ comments, we would like to resubmit the review for publication.
Changes are underlined in the revised version of the manuscript.
Yours Sincerely,
Giovanni Monteleone (on behalf of all the authors)
Department of Systems Medicine
University of Rome Tor Vergata, Via Montpellier, 1, 00133 Rome, Italy
Phone +39.06.72596158
Fax +39.06.72596158
Email: gi.monteleone@med.uniorma2.it
Reviewer 2:
It is an in depth analysis of the role of the Aryl Hydrocarbon Receptor [AHR] [a] in controlling the "healthy" intestine inflammation, [b] as a "treatment" in experimental colitis in mice and [c] in the early steps in IBD patients. This is a review successfully gathered all information regarding the role of AHR in experimental and clinical colitis. This review is well structured and contains the up-to date literature.
There is one table, summarizing the current clinical research and one, well designed figure depicting the way of action.
My unique suggestion is that the authors should divide chapter 3, on "the role of AHR in the maintain ace of gut homeostasis", into sub-part, according for example it effect on the different cell types; the only reason being to be easily understandable, since it is too long - 4 page, including the figure.
Response: We would like to thank the reviewer for her/his positive evaluation and helpful suggestion. We have divided paragraph 3 into sub-paragraphs as suggested by the reviewer.
Round 2
Reviewer 1 Report
Comments and Suggestions for Authors
The ms is now suitable for publication